# Predicting Functions of Uncharacterized Human Proteins: From Canonical to Proteoforms

**DOI:** 10.3390/genes11060677

**Published:** 2020-06-21

**Authors:** Ekaterina Poverennaya, Olga Kiseleva, Anastasia Romanova, Mikhail Pyatnitskiy

**Affiliations:** 1Department of Bioinformatics, Institute of Biomedical Chemistry, 119121 Moscow, Russia; olly.kiseleva@gmail.com (O.K.); romanova.aa@phystech.edu (A.R.); mpyat@bioinformatics.ru (M.P.); 2Institute of Environmental and Agricultural Biology (X-BIO),Tyumen State University, 625003 Tyumen, Russia; 3Faculty of Biological and Medical Physics, Moscow Institute of Physics and Technology, Dolgoprudny, 141701 Moscow, Russia; 4Department of Molecular Biology and Genetics, Federal Research and Clinical Center of Physical-Chemical Medicine of Federal Medical Biological Agency, 119435 Moscow, Russia

**Keywords:** protein coding genes, function annotation, Gene Ontology, protein–protein interaction, splice form, proteoform, uPE1 proteins, human interactome, AP-MS, BioPlex

## Abstract

Despite tremendous efforts in genomics, transcriptomics, and proteomics communities, there is still no comprehensive data about the exact number of protein-coding genes, translated proteoforms, and their function. In addition, by now, we lack functional annotation for 1193 genes, where expression was confirmed at the proteomic level (uPE1 proteins). We re-analyzed results of AP-MS experiments from the BioPlex 2.0 database to predict functions of uPE1 proteins and their splice forms. By building a protein–protein interaction network for 12 ths. identified proteins encoded by 11 ths. genes, we were able to predict Gene Ontology categories for a total of 387 uPE1 genes. We predicted different functions for canonical and alternatively spliced forms for four uPE1 genes. In total, functional differences were revealed for 62 proteoforms encoded by 31 genes. Based on these results, it can be carefully concluded that the dynamics and versatility of the interactome is ensured by changing the dominant splice form. Overall, we propose that analysis of large-scale AP-MS experiments performed for various cell lines and under various conditions is a key to understanding the full potential of genes role in cellular processes.

## 1. Introduction

Prior to the start of the “Human Genome” international project, it was assumed that our genome contains ca. 100 thousand protein-coding genes (PCGs, [1]), which determine the complexity of the human body. In 2004, the draft of the human genome decreased the catalog of genes 5-fold compared to previous estimates [2]. The diversity of living systems is achieved by the factors of post-genome heterogeneity—thus, up to 100 distinct protein variants can be translated from the same gene [3,4]. Despite tremendous efforts of the genomics community, the challenge of identifying all human PCGs still confronts us: data in neXtProt and UniProt are being constantly updated [5,6]. The ongoing project “Human Proteome” [7] is aimed to find the answer for a fundamental question: How many protein-coding genes do we have and what are their roles in cellular processes?

Since the start of the Human Proteome Project, significant progress has been made in registering at the proteomic level more than 90% protein-coding genes [8]. However, despite the existence of many protein function annotation methods based on various data types (sequences, protein interactions, co-expression, and etc. [9,10,11,12,13,14]) for a reasonable fraction of detected proteins (13%), there is no evidence of their function. To fill this gap, in 2017, C-HPP (Chromosome-Centric Human Proteome Project) announced the neXt-CP50 challenge [15], which is aimed at functional annotation of uncharacterized proteins that were previously detected at the proteomic level—so called uPE1 proteins [15,16]. In three years, the number of uPE1 proteins reduced from 13% to 6%.

Proteins entangle in a dynamic web of interactions to mechanize their functions through delicate interplay with their partners. Protein–protein interactions (PPIs) are subtle and dynamic matter, adjusting themselves to the changing environment [17]. Interactomics provide panoramic insight into the machinery of complex biological processes and proteins involved in disease development [18]. Thus, interactomics is one of the main suppliers of protein functions annotation, where at present, no functional data are available for 1193 uPE1 proteins.

In the vast majority of cases, the term “protein” includes all the proteoforms encoded by a single gene. Due to complexity of proteoform identification, they are often neglected in interactome analysis and function annotation. However, since proteoforms can differ in their function, in the context of postgenomic studies, the usage of the term “master form” [4,19] is more correct when it is impossible to identify specific proteoforms.

During the last two decades, significant progress has been achieved in building human interactome maps [20]. Such maps casted light upon fine-tuning of protein characteristics [21], evolution [22], and disease [23,24]. Over the years, several elegant tools of PPI exploration emerged [25,26], including affinity purification coupled to mass spectrometry (AP-MS) [27]. The idea of the latter method is to capture proteins of interest from a solution using immobilized “bait” proteins and to identify affinity-purified targets with LC-MS (“preys”).

The major advantage of AP-MS over other interactomic techniques is its environmental compatibility: natural heterogenic content (including proteoforms) of purified proteins remains stable and adequately reflects the majority of PPIs inside the cell. Unfortunately, the step of cell lysis leads to the loss of fragile temporal and spatial interactions [28], which maintain communication between molecules and “stick together” the whole interactome [29].

Experimental and processing imperfection results in false positive results, biases, and errors flooding interactomic databases [30]. Although availability of omics datasets is growing [31], labor-intensive interactomic and proteomic datasets are still sparingly cited and re-analyzed, which hampers the estimation of their accuracy and reproducibility and nullifies their impact on landscape of biological knowledge [32,33].

Thus, as was shown earlier [34,35,36,37], interactomics may gain novel knowledge from already existing datasets. In this study, we re-analyzed AP-MS data obtained in the largest high-throughput interactomics project BioPlex 2.0 [38,39] to explore co-occurrence of proteoforms (canonical and alternatively spliced variants) for functional annotation previously uncharacterized genes and proteins.

## 2. Materials and Methods

### 2.1. Gene Sets

The list of human PCGs was obtained from neXtProt v. 11-2019 [6]. Additional information was obtained on the number of translated splice forms, and genes, which were confirmed at the proteomic level, but still have no functional annotation (uPE1 proteins) [15,16]. The list of baits was extracted from Huttlin et al. [39].

### 2.2. Re-Analysis of MS Data

We collected raw published data from BioPlex 2.0 [39] via individual query for each human gene. Raw MS/MS data were converted into mgf format by MSConvert and processed in a uniform manner by three search engines (X!Tandem, MS-GF+, OMSSA) [40] being a part of SearchGUI (v. 3.3.15) package coupled with PeptideShaker (v. 1.16.40) [41]. Acquired LC-MS/MS data were searched against the human neXtProt library (rel. 2019-01-11) and enriched with CRAPome contaminants [42]. Mass tolerances were set to 10 ppm and 0.5 Da for precursors and fragments, correspondingly. Carbamidomethylation of cysteine residues was set as a fixed modification and oxidation of methionine was allowed as a variable modification. Only highly confident peptides according to target-decoy approach were accepted. For both peptides and proteins, FDR cut-off was set to 1%. At least two detected peptides (and at least one of them to be unique) were required for protein identification.

### 2.3. Building PPI Network

We used matrix-model interpretation of protein–protein interactions where all the proteins occurring in the same purification experiment, either bait–prey or prey–prey pairs, are recognized as connected with each other. The Dice coefficient was utilized to measure the degree of confidence for interaction between proteins. The Dice coefficient for two proteins, *i* and *j*, is defined as:(1)$$D\left(i,j\right)=\frac{2q}{\left(2q+r+s\right)},$$
where *q* denotes the number of times that both proteins appeared in the same purification, *r* denotes the times that only protein *i* occurs, *s* denotes the times that only protein *j* occurs [43]. Bait-prey data from AP-MS experiments were processed using SMAD package [44].

### 2.4. Determination of Optimum Threshold

Results obtained using Dice coefficient were compared with a scheme presented by Hart et al. [45]. The latter approach utilizes hypergeometric distribution to estimate the probability of interaction between two proteins being observed at random given the total number of interactions for each protein. As a golden standard we used the CORUM database, literature-curated set of human protein complexes [46], which is commonly used for evaluating global interactome networks [29,35,47,48]. We utilized the Core Complexes dataset which is essentially free of redundant entries from the latest available CORUM release 3.0. Dataset was restricted to cover only human proteins and the ‘Disease comment’ column was filtered to contain only ‘None’. We considered 2591 proteins present both in CORUM and our datasets and defined the following numbers:TP = number of interactions observed both in CORUM and in our dataset.FP = number of interactions observed in our dataset but not in CORUM.FN = number of interactions observed in CORUM but not in our dataset.

The performance of both Dice and Hart methods was assessed by computing the corresponding F_1_ score, a measure of test accuracy defined as a weighted average of the precision and recall:(2)$$Precision=\;\frac{TP}{\left(TP+FP\right)}\;Recall=\;\frac{TP}{\left(TP+FN\right)}\;F_{1}=\;\frac{2\times Precision\times Recall}{\left(Precision+Recall\right)}=\frac{2\times TP}{\left(2\times TP+FP+FN\right)}.$$

The dependence between threshold quantile and *F*_1_ score for Hart and Dice indices is presented in Appendix A. We found that overall, Dice score performed much better and achieved *F_max_* = 0.145 (F_1_ score providing the maximum performance, Dice threshold = 0.371), while Hart showed *F_max_* equal to 0.086.

### 2.5. GO-Annotations for Characterized Proteins

Each protein from the PPI network and not included in the uPE1 list (total 9561 proteins including 9558 having “reviewed” status) was annotated with its GO-terms from Uniprot using ViSEAGO package [49]. We used the “2020-03” GO release and “2020_01” UniProt release.

In the present work, we pursue the idea of minimizing the total number of GO-terms in order to avoid unreliable results and reduce the number of spurious protein function predictions. The filtering procedure consisted of three steps. At first, we limited our analysis to GO-terms which had experimental evidence codes: Experiment (EXP), Direct Assay (IDA), Physical Interaction (IPI), Mutant Phenotype (IMP), Genetic Interaction (IGI), and Expression Pattern (IEP). Initially there were 1034 distinct GO-terms for Cellular Component sub-ontology (CC), 2180 distinct terms for Molecular Function (MF), and 5723 distinct terms for Biological Process (BP) sub-ontology, while the total number of terms for all proteins was 17,721, 17,419, 22,862 for CC, MF, and BP respectively.

As the second step, in order to remove too rare terms, we filtered out GO-terms which were supported by less than 10 proteins. This allowed further reduction of the dataset to 206 CC terms, 209 MF terms, and 424 BP terms.

Finally, we filtered out high-level (i.e., too specific) GO-terms in order to predict low-level (more general) terms more reliably. We removed GO-terms which had higher than 10-th level for CC and BP, and 6-th level for MF. These thresholds were selected to remove approximately half of the GO-terms from each sub-ontology. Finally, there were 101 CC terms, 107 MF terms, and 273 BP terms.

### 2.6. Prediction of Unknown Protein Functions

Protein function prediction was performed using COSNet [50], Hopfield-based cost sensitive neural network algorithm for learning node labels in partially labeled graphs. Briefly, the algorithm consists of three steps: (1) unlabeled nodes are randomly assigned with positive and negative labels in the same proportion as labeled nodes; (2) optimal parameters of the labeled subnetwork are found through optimizing the F-score criterion; (3) the regularized dynamics with learned parameters on the unlabeled nodes subnetwork is simulated until an equilibrium state is reached. Details are given in [50]. COSNet is specifically designed to work with biological networks where labelings are highly unbalanced—in most cases, the number of proteins with specific function (positive class) is much smaller compared to overall number of proteins which do not possess this function (negative class). An important feature of the algorithm is that the requirement to output function prediction for each protein is optional—if the data are insufficient for reliable answer, the function will remain unknown.

The COSNET algorithm takes as input graph weights reflecting similarity between nodes. We used the Dice score values as graph weights. The second parameter of the algorithm is regularization cost. We performed a grid search for different cost values in the range from 10 to 10^−10^, optimizing F_1_ score over GO categories which had experimental evidence codes, i.e., known protein functions. For each run, randomly selected 80% genes with annotated GO-category were used as a training set for COSNET and the remaining 20% were used as a test set. The obtained F_1_ scores for all GO-categories were averaged for each value of regularization cost and the training and test set were unrelated at each iteration. The optimal cost value was established at the level of 10^−5^ for all GO categories.

### 2.7. Analysis of Interactome Profiles

The isolation of functional clusters for the identification of common partners was carried out by pairwise comparison of the obtained interactomic profiles (list of interacting partners) for all proteins within the network. The number of common partners was used to calculate the measure of the relationship between two proteins. The proteins having more than 50% common partners were combined into one group. Each group was used as the ‘target set’ in the GOrilla web service [51] for GO enrichment [52] A complete set of 20,223 human proteins was used as a reference. The results with significance level *p* < 10^−7^ were processed. The visualization of pairwise interactions was carried out using Cytoscape software (v.3.7) [53].

### 2.8. Software Implementation of Algorithms

Calculations and graphics were performed in Python 3.6 and R 3.6 [54] with packages igraph [55], ComplexHeatmap [56], graphlayouts [57], and threejs [58].

## 3. Results

### 3.1. Identification of Proteins

Interactome database BioPlex [38,39] is a developing project on building a human interactome by sequential annotation of human genes using the affinity purification coupled with mass spectrometry approach (AP-MS, [59]). We downloaded and processed 11,532 MS-files from the BioPlex 2.0 database [39]. These MS data corresponded to the results of AP-MS experiments for 5766 genes (“baits”). According to the Human Proteome Project terminology, 252 genes of this list belong to the so-called uPE1 proteins—the expression of such genes is confirmed at the proteome level, but the function is still unknown [15].

A total of 12,444 sequences encoded by 11,308 genes were detected, among which 550 uPE1 proteins were identified (Figure 1a). It is noteworthy that 27 genes encoding uPE1 proteins were identified in both the canonical and splice form (Figure 1b), one gene was identified only as the splice form, and for 179 of the 476 genes, both the canonical form and master form were identified [4,19], or only master form—when specific amino acid sequence is uncertain. A total of 1156 splice forms were identified for 1076 genes, with 671 genes not represented by a uniquely defined canonical form, 25 genes represented only by splice form, and one gene—by several splice forms (Figure 1b, Appendix A). It should be noted that among the “baits” represented initially by canonical sequences, for 37.6% of genes, there is no information about protein sequences resulting from alternative splicing.

The authors of the BioPlex project constructed HA-FLAG-tagged open reading frames (ORFs). The description of the protein sequence is available for 5037 baits. According to the ORFEOME database [60], 1424 sequences were represented by splice forms. Among these splice forms 70 belong to uPE1 proteins. However, only 204 of the 252 uPE1 proteins were identified as potential preys. In total, 3698 baits were also detected as preys: 2237 baits were presented in the original sequence, for 1440 cases, it was impossible to determine the specific form of the amino acid sequence (i.e., the master form), and for 22—another amino acid sequence was detected (Appendix A). It should be noted that among the baits represented initially by the canonical sequence for 37.6% of genes, there is no information about protein sequences resulted from alternative splicing.

Thus, at the protein identification level, it can be noted that in half of the cases, there is a prevalence of alternatively spliced forms. Obtained proteomic data on the total identification reflect the fact that only 50–60% of human genes are expressed per tissue [61].

### 3.2. Human PPI Network

The graph of PPIs was built using the Dice scoring scheme [62]. Two proteins were considered to interact if their Dice score was greater than the predefined threshold. In order to select optimum cut-off value, we varied the threshold and compared predictions with the golden standard—CORUM database, a manually curated repository of experimentally characterized protein complexes [46]. The optimum Dice threshold for CORUM proteins according to the F_1_ score was determined to be equal to 0.371 and this value was used to build the global PPI network. We also evaluated another metric based on hypergeometric distribution presented in [45] but found that Dice scoring was performing considerably better (see Materials and Methods, Appendix A). When constructing the PPI graph, we accepted master proteins as a canonical form.

The final PPI network consisted of 9967 vertices (proteins) connected by 287,474 interactions with 8686 proteins forming a single giant component. Each protein on average interacted with 5 partners (median value). Complete network structure in graphml format is available in the Supplementary.

Among the proteins with detected PPIs, 3630 (from 5037 identified) belonged to the baits’ genes. At the same time, 15 baits’ genes were represented by an alternative sequence, for other cases, due to accepted master form as canonical, we cannot be sure. In total, the PPI network contained 1026 splice forms, while only for 805 of 963 genes, there was also a canonical version, and 57 genes were represented by several splice forms. PPIs were obtained for 406 of 550 identified uPE1 proteins (corresponding to 378 genes of 476 detected), 26 genes were represented by both canonical and splice forms, and another 352—only by canonical form.

### 3.3. GO Category Prediction for uPE1 Proteins—Biological Processes

The COSNet algorithm predicted at least one biological process to 256 uPE1 proteins out of 406 and the total number of distinct GO BP-terms was 204. The overall binary matrix with predictions is available in Appendix A.

Since it is impossible to fully plot the obtained data due to their high dimensionality, we tried to present the predicted protein functions from a bird-eye view. We resorted to data filtering by removing both rare GO categories and proteins annotated with a small number of categories (in both cases five or less proteins/categories). Further, for visualization purposes, the total number of GO categories was additionally reduced. We collapsed GO categories into 15 clusters using the GOSemSim package [48]. Among GO categories that fell into one cluster, one category with the lowest level was selected, i.e., least specific. As a result of removing low-annotated proteins, rare and high-level GO-terms the original binary 256 × 204 prediction matrix (proteins vs. GO-terms) was reduced to a 127 × 15 matrix. We would like to stress that all filtering and clustering of proteins and GO categories was performed only to give a big picture of results for visualization purposes, while all actual predictions are given in Appendix A.

Since each protein is assigned to a set of GO BP-terms, we visualized the intersection between these sets as an Upset plot (Figure 2), which can be viewed as a generalization of a Venn diagram to deal with more than three sets. Each row in the plot represents a single GO-term and each column represents proteins which were predicted to have one or more terms (shown by black dots and names of corresponding proteins at the top).

The right side of the graph shows the number of proteins annotated with the “biological process” according to the prediction. Five categories were predicted more frequently than others: protein localization (GO:0008104), heart development (GO:000750), G2/M transition of mitotic cell cycle (GO:0000086), cilium movement (GO:0003341), and microtubule cytoskeleton organization (GO:0000226). Taken together, these categories contain 68 proteins on average, while the remaining 11 categories contain an average of 12 proteins each. Moreover, a combination of these five categories was assigned for the largest number of proteins (24 proteins). For 10 proteins (A2RUT3, A6PVS8, Q6IPT2, Q6IPT2-2, Q7Z5L2, Q8N3J3, Q8N3J3-3, Q8TD91, Q8TD91-2, Q9HA90), a combination of biological processes including proteolysis (GO:0006508), cognition (GO:0050890), and regulation of cell migration (GO:0030334) was predicted. For 19 proteins (rightmost column), other BP categories were predicted (not shown on the plot).

Next, we checked if various biological processes for canonical protein form and for any of its proteoforms were predicted. Two such cases were found. Compared to its canonical variant Q8IYS2, splice form Q8IYS2-2 was assigned 14 additional BP-terms related to regulation of GTPase activity, response to lipopolysaccharide and Golgi to plasma membrane protein transport. In addition, for the proteoform O43149-3, participation in the biological process GO:0016485 “protein processing” was predicted, which was not observed for the canonical form O43149.

### 3.4. GO Category Prediction for uPE1 Proteins—Molecular Functions

We were able to predict at least one molecular function to 380 proteins out of 406 (total number of distinct GO MF-terms was 66), binary matrix with predictions is available in Appendix A. At the same time, the apparent molecular function “protein binding” (GO:0005515) was predicted for 372 proteins, which was not considered in further analysis.

Similar to the processing of BP predictions, the dimensionality of the data was reduced. Rare GO categories and proteins annotated with a small number of categories (five or less) were removed. After this filtration, 32 GO categories were predicted for 60 proteins. Moreover, 36 proteins had functional profiles identical to those already existing. The following biological processes were predicted for 58 proteins: growth factor binding (GO:0019838), kinesin binding (GO:0019894), and GTP-dependent protein binding (GO:0030742). These categories were not taken into account in further visualization.

Since the total number of proteins and MF categories was less than in the case of BP prediction, we used a more descriptive type of plot [63] to visualize the results (Figure 3). As before, the remaining GO categories were grouped into 10 clusters, and the category with the lowest level, i.e., the least specific, was selected as a representative of the cluster. The top-3 most-represented categories included SH2 domain binding (GO:0042169), ubiquitin-like protein ligase binding (GO:0044389), and ephrin receptor binding (GO:0046875).

Four cases of prediction of different molecular functions for canonical protein and its proteoforms were discovered. For canonical protein O43149, ubiquitin-protein transferase activity (GO:0004842) was predicted, while splice form O43149-3 presumably has peptidase activity (GO:0008233). Two new molecular functions “identical protein binding” (GO:0042802) and “protein homodimerization activity” (GO:0042803), which are absent in the canonical form, were predicted for the proteoform O60941-3. Proteins Q8IYS2 and Q8IYS2-2 supposedly possess protein binding (GO:0005515), moreover, we predicted phospholipid binding (GO:0005543) for the canonical form, and actin binding (GO:0003779) for the splice variant. Finally, 13 different molecular functions were predicted for the Q96SK2 and Q96SK2-2 proteins, but the canonical version presumably has protein-macromolecule adapter activity (GO:0030674).

### 3.5. GO Category Prediction for uPE1 Proteins—Cellular Components

The COSNet algorithm was able to predict at least one cellular component for 295 of 406 uPE1 proteins, while the total number of distinct CC-terms was 71. The binary matrix with predictions is available in the Supplementary. The top-3 most represented predicted cellular components included cytosol (GO:0005829, 86 proteins), plasma membrane (GO:0005886, 84 proteins), and cytoplasm (GO:0005737, 83 proteins). After filtering rare GO categories and poorly annotated proteins, the prediction matrix was reduced to 93 proteins and 41 CC categories. Visualization of the distribution of proteins by cellular components is presented in Figure 4.

Five cases of prediction of different cellular components for canonical protein forms and their proteoforms were identified. While cellular localization was not predicted for the O43149 protein, the localization in membrane raft (GO:0045121) and extracellular space (GO:0005615) are presumably specific for its splice form O43149-3. Of note, for this protein and its proteoforms, a difference in the predicted biological processes and molecular functions was also stated. Protein Q8IYS2 has putative localization in lipid droplet GO:0005811, while its splice form Q8IYS2-2 is possibly localized in cytosol (GO:0005829). As in the previous case, earlier for this pair, we noted a difference in biological processes and molecular functions. The localization in the nucleus (GO:0005634) was predicted for the proteoform O60941-3, distinguishing it from the canonical form. For this pair, different molecular functions were previously predicted. For the Q96HA4 and Q96HA4-4 proteins, 13 coinciding CC terms were predicted, and for the splice form, additional localization in cytosol (GO:0005829) was predicted. Finally, 14 coincident localizations were predicted for the Q96SK2 and Q96SK2-2 proteins, but the canonical version is also supposedly localized in the Golgi apparatus (GO:0005794), while the splice form is likely to be localized in mitochondria (GO:0005739). We also predicted different molecular functions for this protein pair.

### 3.6. Differences of Splice-Forms Interactomic Profiles

With the uPE1 protein example, it was shown that proteoforms (splice forms) encoded by a single gene can significantly differ in their functions. We attempted to visualize the interactions between the splice-forms and corresponding canonical proteins. From the complete PPI network, we selected the subgraph containing only canonical proteins having at least one corresponding proteoform (*n* = 1833). We also included hub proteins into the subgraph to preserve the original network architecture as much as possible. The largest connected component of the obtained subgraph containing 14 splice forms (upper level) and 294 canonical proteins (lower level) is presented on Figure 5. The graph is visualized in 3D with the upper level containing splice forms and lower level containing canonical forms. Hub proteins with more than 150 interacting partners are marked with yellow. Of interest we found that isoform 6 of Q9UM54 (myosin-VI) is predicted as a hub protein with a total 556 interactions. The PPI differences between Q9UM54-6 and its canonical form Q9UM54 is about 10% and since this protein is a hub, such discrepancies may be crucial for its cellular role. In total, for 8 out of 14 splice forms, we observed the differences on PPIs with canonical form more than 50%.

For estimation of the total number of splice forms differing by function from canonical form, we compared interactome profiles for a set of proteins encoded by the same gene (62) with more than 10 PPIs. It was shown that, in two thirds of the cases, there are differences in the list of PPIs by more than 50%, and for 24 cases—by more than 90% (Table 1). At 50–70% similarity of interactome profiles for the canonical and splice forms, we observed a similar amount of the corresponding PPIs (CV ~20–40% in comparison with 80–90% in other groups).

Based on a comparison of interactomic profiles (see Materials and Methods), functional clusters were identified to describe the roles of canonical and spliced forms in cellular processes. For the analysis, we considered only cases where the coincidence of the interactomic profiles (consisting of at least 10 proteins) amounted to more than 50%. A total of 157 functional components were isolated, including one macro-component, two mid-components, 48 mini-components, 69 micro-components of 3-7 proteins, and 37 two-proteins components (Appendix A).

Canonical proteins and splice forms encoded by 31 genes fell into various annotation functional clusters (Appendix A and Appendix A). Out of public interactome resources, only the IntAct database [64] holds PPI for splice forms. There is information about different PPI for proteins pair (the same canonical and splice form) encoded by 18 of 31 genes. However, only for 5 cases, the number of interactions for both canonical and splice form was more than 10. Unfortunately, there are not enough data on PPIs for GO annotation and their comparison with our data.

Among 31 genes for which we observe different functional clusters for canonical and splice form, two genes belong to uPE1 encoding Q96HA4/Q96HA4-4 and Q96SK2/Q96SK2-2, for which different GO categories were predicted.

### 3.7. Impact of Data Sources

We attempted to quantify the effect of using another type of PPI data. For this purpose, we utilized the largest binary PPI network obtained by Y2H [26]. The characteristic features of Y2H data include definite protein sequence as opposed to AP-MS and ignoring of the cell environment. Thus, only canonical variants of proteins could be considered because there is no information about splice form despite Y2H proteins being constructed on the basis of ORFEOME.

The Y2H network consisted of 51,763 interactions between 8099 proteins, and 7978 proteins formed a single giant component. Further analysis was limited to 4156 proteins present in both Y2H and AP-MS networks. For these proteins, we retrieved experimental GO BP terms and performed filtering of low-annotated proteins and rare/specific GO-terms as described in the Material and Methods section. The obtained network included 1335 proteins which were further split into train and test in the 80:20 proportion. For each protein from the test sample, we computed the average Wang similarity between sets of predicted and true GO terms [65]. Prediction using the AP-MS network showed superior average similarity to ground-truth GO sets compared to Y2H (0.158 ± 0.006 vs. 0.143 ± 0.005, mean ± 95% CI, Wilcoxon *p* < 0.001). Thus, we conclude that usage of AP-MS networks may be beneficial for protein function prediction compared to Y2H datasets, although thorough exploration of this question is beyond the scope of this study.

### 3.8. Comparison with UniProt Predictions

For 574 out of 1193 uPE1 proteins, there is data on their possible GO categories in UniProt (on average 1.8 terms per protein). Biological processes and molecular functions are known for 30 and 43 uPE1 proteins, respectively, and for the half of uPE1 proteins (550), there are data on cellular localization. In total, we were able to compare the data for 189 uPE1 proteins, where for half of the proteins (92), the predictions coincided with at least one term within two levels. The obtained assessment is rough since different sources of prediction of GO categories were used, and the data were not found to be reliable according to neXtProt, which is the main human proteome database in the frame of the Human Proteome Project.

## 4. Discussion

Accumulation of interactome data contributes to the identification of the role of proteins in cellular processes. Large-scale experiments provide a basis for formulating hypotheses about the functioning of the cell with minimal biological [66] and batch effects [67]. The BioPlex project is the largest database of AP-MS experiments, containing novel experimental proofs of interaction and possible functions for several previously unannotated proteins. The resulting data array is an important component in the development of bioinformatic algorithms for predicting and analyzing PPIs [35].

Through analyzing the results of AP-MS experiments from the BioPlex 2.0 project, we were able to predict GO categories for a total 391 proteins with unknown functions. Of these proteins, we predicted biological processes for 256 proteins, molecular functions for 380 proteins, and cellular components for 295 proteins. All types of GO sub-ontologies were predicted for 219 proteins (Figure 6).

In the context of uPE1 proteins, annotation structure-based prediction and annotation pipeline, combining I-TASSER and COFACTOR algorithms [68], were considered. However, the corresponding analysis was limited to only several uPE1 proteins.

An interesting issue is the prediction of MF based on the PPI network. Indeed, it seems reasonable that proteins with similar molecular function should rather have similar sequence and structure than to be neighbors on the PPI network. However, there is evidence that some information about MF is encoded in the PPI structure. For example, Kulmanov et al. [9] utilized deep learning to learn features from both protein sequences and the PPI network to predict MF, BP, and CC terms. They reported that the model, which relies only on protein sequences, does not achieve improved performance compared to the reference method (BLAST). Several related papers devoted to deep learning PPI-based methods report better or similar model performance for MF compared to BP and CC terms [69,70]. Furthermore, the keyword analysis of CAFA3 participating methods showed that protein interaction is equally important across all 3 GO ontologies [10]. Hence, it may be concluded that information about protein molecular functions is also encoded in the PPI networks structure.

One of the main advantages of the AP-MS is the ability to identify proteoforms in the results of mass spectrometric analysis, while the identified alternatively spliced proteoforms do not require additional verification—contrary to proteoforms with point mutations or post-translational modifications. Our interest in proteoforms is due to the fact that the functions of protein variants of the same gene may differ dramatically, and most often functional differences are observed in splice forms [21,71].

The choice of alternatively spliced protein variant as the major—canonical—sequence is speculative [6] and is refined constantly with an accumulation of transcriptomic and proteomic data. When determining the function of a protein, the type of proteoform (e.g., splice variant) is often ignored due to the methodological limitations. However, analysis of the AP-MS data allows to suggest the functional role of specific splice forms (proteoforms).

Four genes encoding proteins with unknown function were provided with the information on the different functions of their canonical and alternatively spliced forms. In total, differences were revealed for 62 proteoforms encoded by 31 genes (4% of all genes provided with information on PPIs for canonical and splice form). A modest number of alternative splice forms is due to the fact that in most identifications, a master form was registered, which was further considered as canonical. At the same time, 158 genes were represented only by splice forms. Noteworthy, there is a significant discrepancy between the protein sequences selected as a bait in the AP-MS experiment and actually translated in the cell.

## 5. Conclusions

In this work, we re-analyzed thousands of large-scale AP-MS experiments to predict functions of uPE1 proteins and their splice forms. We were able to predict Gene Ontology categories for a total of 387 uPE1 genes. Four uPE1 genes were provided with the information on the different functions of their canonical and alternatively spliced forms. In total, functional differences were found for 62 proteoforms encoded by 31 genes.

Based on the results, it can be carefully concluded that the dynamics and versatility of the interactome is ensured by changing the dominant splice form. This is consistent with observations on the relationship of specific splice forms and the development of pathologies [23,24]. A similar mechanism can be assumed in the case of other types of proteoforms. Thus, in order to understand the full potential of genes in cellular processes, we suggest one should focus on large-scale AP-MS experiments performed for various cell lines and under various conditions.

## Figures and Tables

**Figure 1 genes-11-00677-f001:**
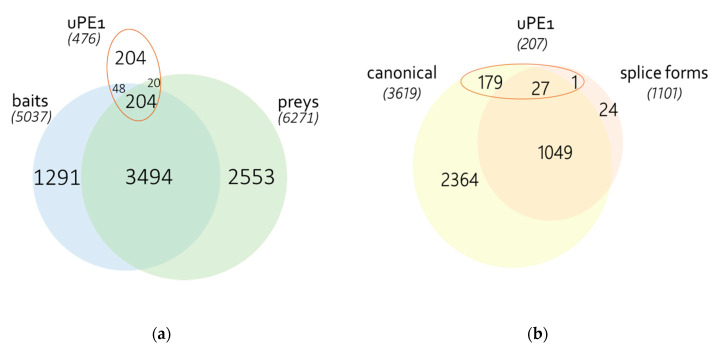
(**a**) Relations between proteins used as preys and baits in BioPlex 2.0 and corresponding uPE1 proteins (**b**) distribution of detected proteoforms encoded by one gene, if alternative splicing is known. Master form is excluded.

**Figure 2 genes-11-00677-f002:**
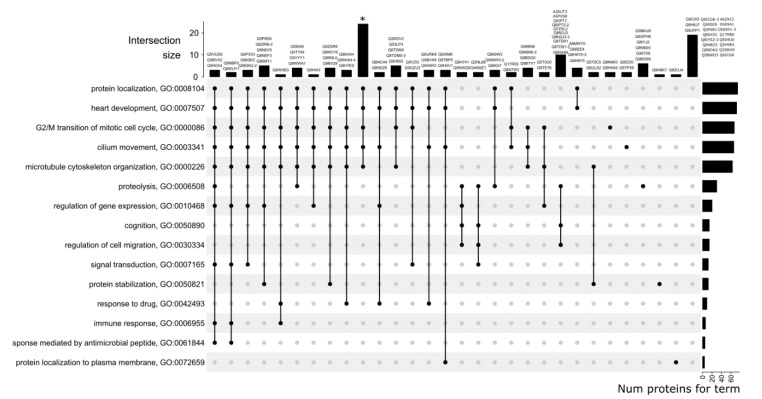
UpSet plot showing predicted biological processes. GO terms for Biological Process (BP) are represented as rows with barplots on the left showing total number of proteins annotated with specific term. Each column denotes proteins annotated with a specific set of GO-terms shown as dots (protein names are given at the top). * denotes the following set of proteins: O00193, O15481, Q53RE8, Q5THK1, Q68CR1, Q6P1M9, Q6P995, Q86X40, Q8IV32, Q8IW50, Q8N7X4, Q8NCJ5, Q8NCT3, Q8NCU4, Q8TBZ0, Q8TDY8, Q8WUB2, Q96BQ5, Q96GQ5, Q9BVG4, Q9H106, Q9H5V9, Q9H910, Q9Y546.

**Figure 3 genes-11-00677-f003:**
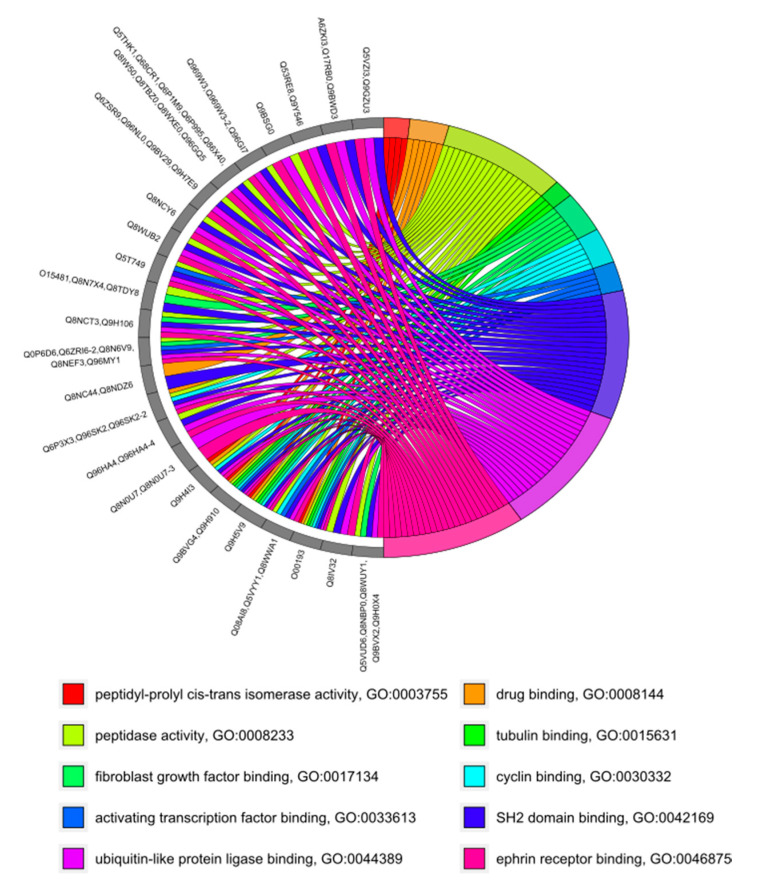
Functional prediction of molecular functions for 60 uPE1 proteins.

**Figure 4 genes-11-00677-f004:**
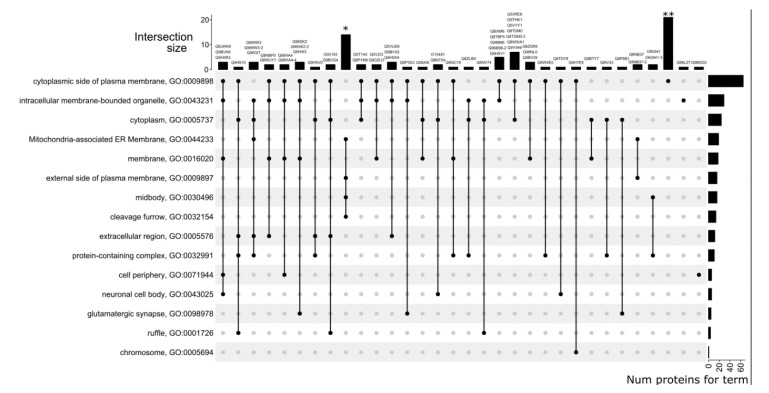
Cellular components predicted for 93 uPE1 proteins. * denotes the following set of proteins: A2RUT3, A6PVS8, Q2NL68, Q4KMZ1, Q6IPT2, Q6IPT2-2, Q7Z5L2, Q8N3J3, Q8N3J3-3, Q8TD91, Q8TD91-2, Q9H741, Q9HA90, Q9NWQ9. ** denotes the following set of proteins: Q0P6D6, Q2M3V2, Q53LP3, Q68CR1, Q6P995, Q6ZRI6-2, Q86X40, Q8IV32, Q8IW50, Q8N6V9, Q8NCJ5, Q8NCT3, Q8NCU4, Q8NEF3, Q8TBZ0, Q8WUB2, Q96BQ5, Q96GQ5, Q96MY1, Q9H106, Q9H693.

**Figure 5 genes-11-00677-f005:**
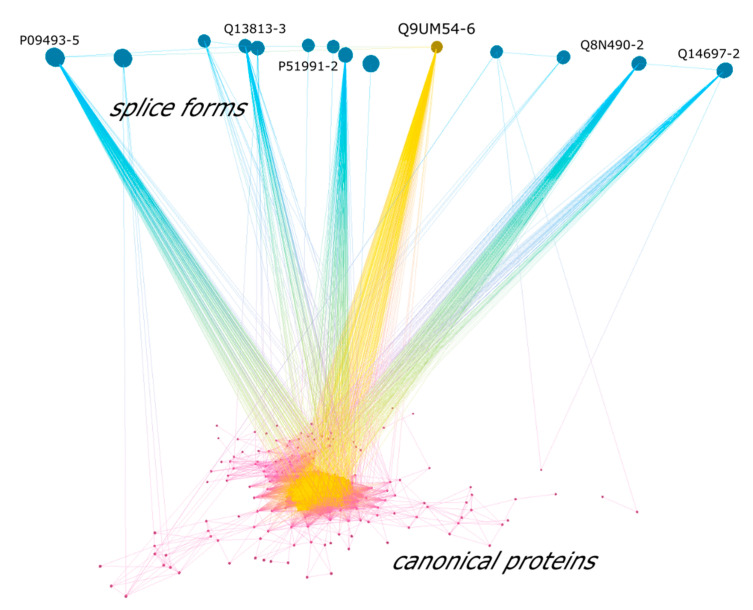
The largest connected component of the obtained PPI network showing canonical proteins (lower level) and their corresponding splice forms (upper level). Hub proteins with more than 150 interacting partners are marked with yellow, including splice form Q9UM54-6 for which we report 10% differences with its canonical form Q9UM54 in terms of PPIs.

**Figure 6 genes-11-00677-f006:**
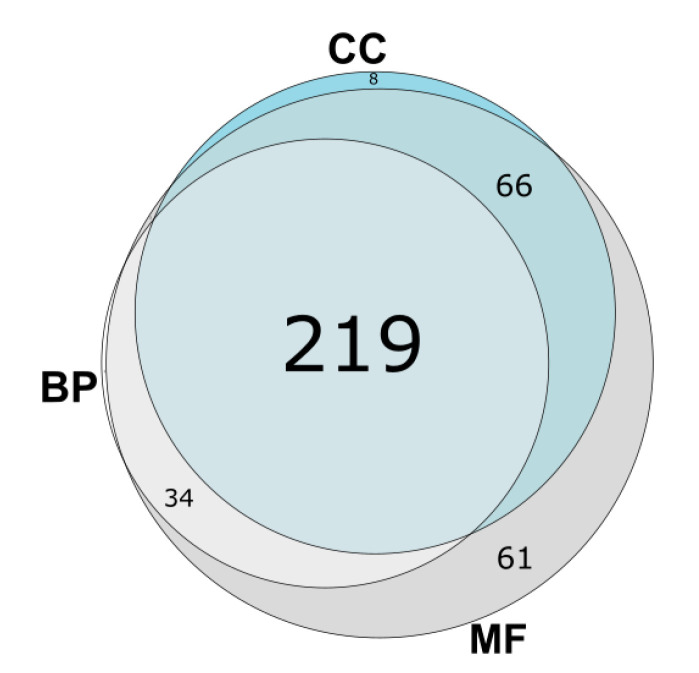
Venn diagram showing distribution of predicted uPE1 protein functions among GO (BP—biological process; MF—molecular function; CC—cellular component).

**Table 1 genes-11-00677-t001:** The differences of interactome profiles for proteoforms encoded by the same gene.

Number of Genes	<50%	≥50%	≥75%	≥90%
62	24	11	3	24

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
