# Peer review of "Predicting Functions of Uncharacterized Human Proteins: From Canonical to Proteoforms"

_genes, 2020, doi:10.3390/genes11060677_

Round 1
Reviewer 1 Report
The paper presents an annotation of a set of non annotated genes and splice variants by means of interactome analysis, leveraging the recently released BioPlex2 data. Per se, the idea of guilty by association is not new and on the overall the authors should better show the novelty of the approach.
Some more issues should be addressed:
- Different interactomes are available: sometimes they are obtained with different techniques (Y2H for example, see the recent Nature 580: 402-408) and give different results (ad discussed in several papers e.g. Trends Biochem. Sci. 42:342-354). Moreover, several databases merge data from different sources obtaining integrative interactomes: what are the advantages and the potential problems in using only BioPlex2? would be the result comparable with different underlying interactomes?
- CORUM is a reference databases of protein complexes (that is obligate, permanent and often stable interactions), not of general protein-protein interactions, that have different scale of stability and that are, for their majorty, transient. The adoption of CORUM should be better justified
- When "training" the parameters, the split of the dataset into training and testing should consider sequence identity in order to obtain unrelated datasets
On the presentation point of view, my suggestions are:
- provide a short explanation of the adopted algorithm, COSNET: I know it is alreadu printed and it has not been developed by the authors, but a short description would improve readability of the paper
- Figures, in particular 1 and 5, should be revised and better explained in the captions. Figs 1 and 5 are obscure to me.
Author Response
Dear Reviewer, we are grateful for your valuable comments. For us it is very important to get critical notes for manuscript’s emendation. The goal of our work is functional annotation of human proteins in context of proteome heterogeneity. Most PPIs investigation is focused on master protein form only, where protein function is сonsidered equivalent to gene function. However one protein-coding gene can be translated into several various proteoforms. Among them splice forms are more sequence-divergent and, as expected, can have more different functions. To estimate possible functions of uncharacterized proteins and splice-forms we used the largest AP-MS dataset available - BioPlex 2.0.
These data were chosen due several reasons:
1) mass spectrometry allows to detect proteoforms in cells;
2) the experiments were performed under uniform conditions - cells, protocols, tools.
For additional minimization of errors we utilized the COSNET method and reduced the number of available GO terms for prediction. Thus overall novelty of our approach includes uPE1 proteins functional annotation combined with the usage of splice forms in PPI network.
Some more issues should be addressed:
- Different interactomes are available: sometimes they are obtained with different techniques (Y2H for example, see the recent Nature 580: 402-408) and give different results (ad discussed in several papers e.g. Trends Biochem. Sci. 42:342-354). Moreover, several databases merge data from different sources obtaining integrative interactomes: what are the advantages and the potential problems in using only BioPlex2? would be the result comparable with different underlying interactomes?
Answer:
Thanks for the important remark. Unfortunately, interactomic data obtained in various ways do overlap by only about 20%. This is evidenced by the results of a comparison of interactome resources and research, mentioned, in particular, in the article by Luck et al., 2017 (Trends Biochem. Sci. 42: 342-354). The reasons for this contradiction lie both in the use of different methods with different limitations, and in the type of biomaterial under study. In other words, the proteome is heterogeneous and dynamic, and there is no single ideal method that can unambiguously define the whole variety of protein interactions.
For our task, we selected BioPlex data, since the AP-MS method allows the identification of proteoforms. Moreover, when using AP-MS, the remaining proteins (except for the bait protein) are in native concentrations. It is also important to note that BioPlex is one of the largest accumulation of PPIs data obtained in a unified manner (one method, one tool base, one biomaterial). Of course, this method is not without drawbacks. For example, a bait protein has increased expression, and a tag can alter the conformation of a protein. Moreover, exploration of a different type of biomaterial will provide different results.
However, our message is that the processing of large-scale AP-MS data taking into account proteoforms allows us to look a little more broadly at the PPI network and the functional annotation of proteins encoded by single gene. New large-scale data, performed with high reliability, as in Luck et al., 2020 (Nature 580: 402-408), allow to improve the training set and continue to study binary interactions. Within the framework of the 3rd release of BioPlex, where it is announced that AP-MS experiments for ~10000 genes are performed, the data obtained can be expanded - as with similar projects on a different type of biomaterial and in other conditions.
We added more explanations to the text for better understanding, as well as Sub-section 3.7 on the analysis of the coincidence of functional annotations for uPE1 proteins by analyzing the Y2H data obtained by Luck et al., 2020 (Nature 580: 402-408).
- CORUM is a reference databases of protein complexes (that is obligate, permanent and often stable interactions), not of general protein-protein interactions, that hav e different scale of stability and that are, for their majorty, transient. The adoption of CORUM should be better justified
Answer:
Thanks for the comment. In the manuscript we re-analyzed results of AP-MS experiments from BioPlex (biophysical interactions of ORFeome-based complexes). The AP-MS technology aims at determining components within multi-protein complexes. In the original paper Huttlin et al state that “BioPlex accurately depicts known complexes, attaining 80-100% coverage for most CORUM complexes” (Huttlin et al, Cell, 2015). Also the adoption of CORUM as a golden standard also have been widely utilized for evaluating global interactome networks - for example in Hein et al (Cell, 2015), Drew et al (Mol Syst Biol, 2017), Scott et al (Proteomics, 2015; and Mol Syst Biol 2017). We have added the corresponding references to the manuscript.
- When "training" the parameters, the split of the dataset into training and testing should consider sequence identity in order to obtain unrelated datasets
Answer:
An important parameter of the COSNET algorithm is regularization cost. For each GO category with experimental evidence (i.e. known protein functions, true answers) and for various values of regularization cost we utilized 80% of genes annotated with current GO term as input to COSNET and tested on remaining 20% genes. The agreement between COSNET prediction and ground truth was expressed via F1 score. The obtained F1 scores for all GO-categories were averaged for each value of regularization cost. At each iteration the dataset was randomly split as 80:20, thus ensuring that train and test samples were completely unrelated. We clarified this issue in the resubmitted manuscript version.
On the presentation point of view, my suggestions are:
- provide a short explanation of the adopted algorithm, COSNET: I know it is alreadu printed and it has not been developed by the authors, but a short description would improve readability of the paper
Answer:
Thank you for this comment, we have extended the Methods section.
- Figures, in particular 1 and 5, should be revised and better explained in the captions. Figs 1 and 5 are obscure to me.
Answer:
Thank you for pointing it out. We have simplified Figure 1 to make it more readable. We also reformulated the description of Figure 5 and corresponding section in the main text to make it more clear.
Reviewer 2 Report
In this manuscript, authors conducted a network analysis based prediction of functions of proteoforms (especially focusing on uPE1 proteins), using the PPI information obtained from AP-MS experiments. The study is timely as the topic of protein function prediction is critical. The general prediction approach used here is not completely original in terms of methodology but the authors set the novelty on discussing the prediction results of proteoforms. The manuscript is written well in terms of the use of language and in terms of the flow of information from chapter to chapter. However, there are a some critical issues related to the current state of the manuscript such as the missing bits of data analysis and the way the results are discussed. Below, I list my specific concerns/issues:
Major issues:
- There are many protein function prediction methods in the literature, some of which use PPI data as their input in network analysis based modelling approaches. In fact, there are a few protein function predictions methods nearly identical to the methodology used here. Many of these similar methods are employed to prediction GO-based functions of the human proteome, including the uPE1 proteins. There is no mention of these methods anywhere in the manuscript, no citation and no quantitative comparison in terms of the obtained predictions for uPE1 proteins. The manuscript sounds like the authors are the first to predict the functions of uPE1 proteins. Please provide the necessary citations to the literature and please compare your prediction results to at least one of these prediction methods (at least in discussion). UniProt provides predicted functional annotations to many of its entries (possible some of the upe1 proteins will be among this dataset), these can be used for the comparison, at least for the canonical forms of upe1 proteins. In this comparison, functional annotation statistics for upe1 genes in UniProt should be given, and compared to your predictions (in terms of correspondence/overlap and differences).
- " 2.3. Building PPI network "
The justification of why the PPI network is built from scratch using raw data (instead of directly using well-established PPI data sources such as IntAct, MINT, STRING or others) not given clearly. Is it only related to using the results of AP-MS experiments? Are the results of AP-MS experiments not included in any of the well-known PPI data resources? If there are PPI databases that incorporate AP-MS data, why not filtering the available PPI data in these PPI databases based on AP-MS experiments?
These PPI databases collects, filters, organizes and publishes PPI information (where the data itself is produced by different experimental approaches) at different confidence levels and at different coverage ratios. It should be possible to obtain a refined, high quality dataset from these databases by increasing the confidence thresholds and by selecting certain criteria. Since these databases are using intense data processing pipelines, their data is probably more reliable compared to the one authors have generated by a simple analysis using the Dice coefficient. Even if the authors prefer generating their own PPI dataset for some other reasons, those reasons should be clearly explained in the manuscript.
On top of that, the same function prediction analysis should be done using PPI data obtained from those well-known databases (without the use of AP-MS experiment results) and the results should be compared with each other (at least in terms of coverage) to justify your choice of generating in-house PPI datasets using AP-MS experiments.
- " 2.5. GO-annotations for characterized proteins "
Which protein set is used here? Are they all of the reviewed human protein entries in UniPRotKB/Swiss-Prot? An explanation is required here. It is also required to include the statistics of this dataset, in terms of both the number of protein entries and the number of GO annotations (for BP, CC and MF) for these proteins, and the database versions of the time the data was obtained.
- " Then we removed GO-terms which had higher than 10-th level for CC and BP, and 6-th level for MF"
Why is this done? Go terms with low number of annotated genes/proteins have already been removed from the dataset prior to this. So why this additional filtering? Please explain the manuscript.
- " We divided all GO categories into 15 clusters using the GOSemSim package"
What is the logic behind this clustering and what are the clustering parameters? Why 15 clusters in particular? Please explain the manuscript.
- " 3.4. GO category prediction for uPE1 proteins - molecular functions "
What is the logic behind predicting MF terms based solely on PPI information? It makes sense to use this info to predict BP terms because interacting proteins usually take part in the same process, it also makes sense for CC prediction because protein complexes are formed by permanent interactions between proteins. But the same thing is not true for molecular functions. When there is, for example, an interaction between two proteins where the first one phosphorylates the second one, only the first one possess this molecular function, not the second one. Similarities between proteins in terms of molecular functions are mostly related to their sequence and structure. Calculating the common interaction partners between two proteins and predicting the MF based function based on this information would make sense but the prediction procedure based on COSNet does not sound like that.
- " In total, confident functional differences were 368 found for 62 proteoforms encoded by 31 genes.''
This expression should be backed with a case study. The authors should pick an example from this list and verify their functional difference predictions based on experimental information from the literature. Predictions can be full of errorsand there is no way to know this without any validations.
- Figure 4 is not informative. Please consider changing it to a format similar to Figure 2 or 3.
Minor issues:
- "The performance of Dice and Hart method was assessed by computing their …"
Hart method is not described anywhere, please include its description together with the part where you describe the Dice method.
- "while Hart showed a max F1 score equal to 0.086."
This metric is called Fmax in the literature (the F1-score at the threshold, which provides the maximum performance). Please use this terminology.
- "Each protein was annotated with its corresponding GO-terms from Uniprot using ViSEAGO"
It is not clear why this package is used? GO annotations of UniProt protein entries can directly be downloaded from UniProt or from the QuickGO browser. What is the use of ViSEAGO here?
- " 2.7. Analysis of interactome profiles"
The analysis is not explained clearly. It sounds like this chapter talking about the method used for "3.6. Differences of splice-forms interactomic profiles", but there are sentences that sound irrelevant. For example: " Certain related groups were annotated by Gene Ontology [34] terms using GOrilla web service". What is the meaning of this? Why and how did you do this GO annotation? What is the significance of the results obtained from the GOrilla annotation?
- " Thus, the original binary 256x254 prediction matrix (proteins vs GO-terms) was reduced to 209 a 127x15 matrix.''
Why the number of proteins are reduced as well at the end of GO clustering process? According to the inheritance rule of GO DAG, a gene/protein annotated with a GO term is assumed to be annotated with the parents of that terms. As a result, the number of proteins should have remained the same.
Author Response
Dear Reviewer, thank you for your detailed and helpful comments regarding the presented manuscript. We really appreciate your feedback and have addressed each of your comments below.
Major issues:
- There are many protein function prediction methods in the literature, some of which use PPI data as their input in network analysis based modelling approaches. In fact, there are a few protein function predictions methods nearly identical to the methodology used here. Many of these similar methods are employed to prediction GO-based functions of the human proteome, including the uPE1 proteins. There is no mention of these methods anywhere in the manuscript, no citation and no quantitative comparison in terms of the obtained predictions for uPE1 proteins. The manuscript sounds like the authors are the first to predict the functions of uPE1 proteins. Please provide the necessary citations to the literature and please compare your prediction results to at least one of these prediction methods (at least in discussion). UniProt provides predicted functional annotations to many of its entries (possible some of the upe1 proteins will be among this dataset), these can be used for the comparison, at least for the canonical forms of upe1 proteins. In this comparison, functional annotation statistics for upe1 genes in UniProt should be given, and compared to your predictions (in terms of correspondence/overlap and differences).
Answer:
Thank you for a very important comment. We partly rewrote the introduction and conclusion for a better understanding of the situation with functional annotation of uPE1 proteins. As you have pointed out, there are many methods for determining and predicting the function of proteins. However, on a large scale, the issue of confident functional annotation of proteins was not considered until 2017.
The proteomic community has already faced the need to develop clear criteria for protein detection, as part of the implementation of the international project “Human Proteome” using “missing proteins” as an example. The next-CP challenge launched in 2017 aims to describe the function for all registered proteins and develop strategies for reliable annotation. So, despite the fact that for half (574 of 1193) uPE1 proteins there is an indication of at least one GO category in UniProt, it is recommended to use information from neXtProt as a more updated and reliable resource for the study of human proteins.
We have added a 3.8 sub-section devoted to comparing our predictions for uPE1 proteins with available records in UniProt to the Results and Discussions. The vast majority of annotations for uPE1 proteins in UniProt are listed under GO CC - 550 out of 574, GO BP and GO MF - 30 and 43, respectively. In total, the coincidence between the obtained and known data was about 60% out of 189 possible cases of comparison. This result is encouraging, because due to the difference in methods, tissue specificity, and neglect of proteoforms, the coincidence between interactomic data, one of the main tools for functional annotation, is about 20% (Luck et al., 2017).
- " 2.3. Building PPI network "
The justification of why the PPI network is built from scratch using raw data (instead of directly using well-established PPI data sources such as IntAct, MINT, STRING or others) not given clearly. Is it only related to using the results of AP-MS experiments? Are the results of AP-MS experiments not included in any of the well-known PPI data resources? If there are PPI databases that incorporate AP-MS data, why not filtering the available PPI data in these PPI databases based on AP-MS experiments?
These PPI databases collects, filters, organizes and publishes PPI information (where the data itself is produced by different experimental approaches) at different confidence levels and at different coverage ratios. It should be possible to obtain a refined, high quality dataset from these databases by increasing the confidence thresholds and by selecting certain criteria. Since these databases are using intense data processing pipelines, their data is probably more reliable compared to the one authors have generated by a simple analysis using the Dice coefficient. Even if the authors prefer generating their own PPI dataset for some other reasons, those reasons should be clearly explained in the manuscript.
On top of that, the same function prediction analysis should be done using PPI data obtained from those well-known databases (without the use of AP-MS experiment results) and the results should be compared with each other (at least in terms of coverage) to justify your choice of generating in-house PPI datasets using AP-MS experiments.
Answer:
Thank you for this comment. Unfortunately, interactomic data obtained in various ways do overlap by about 20% according to comparison of PPIs database. This is evidenced by the results of a comparison of interactome resources and research, mentioned, in particular, in the article by Luck et al., 2017 (Trends Biochem. Sci. 42: 342-354). The reasons for this contradiction lie both in the use of different methods with different limitations, and in the type of biomaterial under study. In other words, the proteome is heterogeneous and dynamic, and there is no single ideal method that can unambiguously define the whole variety of protein interactions.
For our task, we selected BioPlex data, since the AP-MS method allows the identification of proteoforms. Moreover, when using AP-MS, the remaining proteins (except for the bait protein) are in native concentrations. It is also important to note that BioPlex is one of the largest accumulation of PPIs data obtained in a unified manner (one method, one tool base, one biomaterial). Of course, this method is not without drawbacks. For example, a bait protein has increased expression, and a tag can alter the conformation of a protein. Moreover, exploration of a different type of biomaterial will provide different results.
However, we attempted to quantify the effect of using another type of PPI data (see 3.7 sub-section). For this purpose we utilized the largest binary PPI network obtained by Y2H [Luck et al, 2020]. We assessed the performance of function prediction analysis using the Y2H network and compared the results with the AP-MS network. The Y2H network consisted of 51,763 interactions between 8099 proteins, and 7978 proteins formed single giant component. Further analysis was limited to 4156 proteins present in both Y2H and AP-MS networks. For these proteins we retrieved experimental GO BP terms and performed filtering of low-annotated proteins and rare/specific GO-terms as described in Materials and Methods. The obtained network included 1335 proteins which were further split in 80:20 as training and test. For each protein from the test sample we computed the average Wang similarity between sets of predicted and true GO terms. Prediction using AP-MS network showed superior average similarity to ground-truth GO sets compared to Y2H (0.158±0.006 vs 0.143±0.005, mean±95% CI, Wilcoxon p<0.001) Thus we can conclude, that usage of AP-MS networks may be beneficial for protein function prediction compared to Y2H datasets, although thorough exploration of this question is beyond the scope of the presented manuscript.
- " 2.5. GO-annotations for characterized proteins "
Which protein set is used here? Are they all of the reviewed human protein entries in UniPRotKB/Swiss-Prot? An explanation is required here. It is also required to include the statistics of this dataset, in terms of both the number of protein entries and the number of GO annotations (for BP, CC and MF) for these proteins, and the database versions of the time the data was obtained.
Answer:
We annotated all proteins from PPI network which were not included in the uPE1 list with GO terms. This protein set consisted of 9561 proteins and all of them except three (Q96S15-2, P21802-10, Q56UN5-6) had the “reviewed” status. We used the “2020-03” GO release and “2020_01” UniProt release. Total number of GO annotations was 22,862, 17,721, and 17,419 for BP, CC and MF respectively. We have added the corresponding statistics to the manuscript.
- " Then we removed GO-terms which had higher than 10-th level for CC and BP, and 6-th level for MF"
Why is this done? Go terms with low number of annotated genes/proteins have already been removed from the dataset prior to this. So why this additional filtering? Please explain the manuscript.
Answer:
Our general intention in the presented manuscript was to avoid overprediction of protein functions by minimizing the total number of possible GO terms. That is the reason why we limited analysis to GO-terms which were annotated with experimental codes. The filtering procedure was done with the same idea in mind - limit the analysis to non-rare and low-level (i.e. more general) GO terms. The filtering was done in two steps. At first we removed rare GO-terms which were supported by less than 10 proteins. Then we removed too high-level GO terms. The reason to do this second step of filtering was that we were unlikely to predict a very specific protein function and better let’s be more reliable by predicting the more general function. Furthermore this additional filtering had little impact compared to the first step. For example, initially there were 5732 distinct BP terms. The removal of rare GO-terms had the largest effect by reducing the number more than ten-fold to 424 terms, while the removal of too specific terms resulted in 273 BP terms. We reformulated the corresponding section in the manuscript.
- " We divided all GO categories into 15 clusters using the GOSemSim package"
What is the logic behind this clustering and what are the clustering parameters? Why 15 clusters in particular? Please explain the manuscript.
Answer:
In the manuscript we attempted to predict protein functions for uPE1 proteins. However the dimensions of obtained predictions (matrices of the form proteins-vs-GO terms) are quite large to plot it as a heatmap: 256x204 for biological processes, 380x66 for molecular functions, 295x71 for cellular components. So in order to visualize the predictions at least from bird-eye view we resorted to various “dimensionality reduction” filtering schemes. One approach to reduce the number of distinct GO terms for plotting purposes is to cluster them together, since some terms may be descendants of others or have other significant semantic similarities. This type of collapsing excessive GO-terms is widely used for visualization of enrichment results, for example in REViGO software. We decided to collapse GO terms into 15 clusters since this number seems to give reasonable detail while not being too large for too sparse picture. We would like to stress out that all this clustering of GO categories was performed only to give a big picture of results for visualization purposes while all actual predictions including all non-clustered GO categories are given as tables in Supplementary. We modified the manuscript to clarify this issue.
- " 3.4. GO category prediction for uPE1 proteins - molecular functions "
What is the logic behind predicting MF terms based solely on PPI information? It makes sense to use this info to predict BP terms because interacting proteins usually take part in the same process, it also makes sense for CC prediction because protein complexes are formed by permanent interactions between proteins. But the same thing is not true for molecular functions. When there is, for example, an interaction between two proteins where the first one phosphorylates the second one, only the first one possess this molecular function, not the second one. Similarities between proteins in terms of molecular functions are mostly related to their sequence and structure. Calculating the common interaction partners between two proteins and predicting the MF based function based on this information would make sense but the prediction procedure based on COSNet does not sound like that.
Answer:
Many thanks for raising this interesting question, we agree with this point of view. Indeed, prediction of MF terms seems to be more reliable based on protein sequence and structure. However there is evidence that some information about MF is encoded in the PPI structure. For example, Kulmanov et al (Bioinformatics, 2018) used deep learning to learn features from both protein sequences and PPI network to predict MF, BP and CC terms. Of interest they report that the model, which relies only on protein sequences does not achieve improved performance compared to the reference method (BLAST). Gligorijević et al (Bioinformatics, 2018) present network fusion method based on multimodal deep autoencoders and report outperforming previous methods for both human and yeast STRING networks. The model cross-validation performance for MF-terms was higher than for CC and BP (Fig 3, Fig S6). In the recent work of Peng et al (Brief Bioinform, 2020), they report that accuracy of DeepMNE-CNN (multi-network embedding algorithm) on human MF-101-300 dataset is comparable to CC-101-300 (0.569 vs 0.594). Hence it may be concluded that information about protein molecular functions is also encoded in the PPI networks. Furthermore, the keyword analysis of CAFA3 participating methods showed that protein interaction is equally important across all 3 GO ontologies (Zhou et al, Genome Biol, 2020). We have modified the manuscript to discuss this interesting issue and also included the warning that this type of prediction may be less reliable and included corresponding references.
- " In total, confident functional differences were 368 found for 62 proteoforms encoded by 31 genes.''
This expression should be backed with a case study. The authors should pick an example from this list and verify their functional difference predictions based on experimental information from the literature. Predictions can be full of errors and there is no way to know this without any validations.
Answer:
We have expanded this part and rewrote Material and Methods and Results sections. We agree that any predictions must be validated. At the same time, in interactomics, where there are many dynamic factors and the “gold standard” method is absent, data comparison does not solve the problem completely.
Nevertheless, we analyzed available data on PPIs for 31 genes mentioned in the IntAct database (see sub-section 3.6). This database is the only source where data on proteoforms are available, but the reliability of their identification is unclear. For 18 of 31 genes, differences in interactomic profiles were observed, but they are not enough for GO enrichment and subsequent comparison with our results.
- Figure 4 is not informative. Please consider changing it to a format similar to Figure 2 or 3.
Answer:
Thanks for pointing this out. Figure 4 has been redrawn in the same format as Figure 2 (UpSet plot).
Minor issues:
- "The performance of Dice and Hart method was assessed by computing their …"
Hart method is not described anywhere, please include its description together with the part where you describe the Dice method.
Answer:
We have included the description of Hart method’s idea in the corresponding place.
- "while Hart showed a max F1 score equal to 0.086."
This metric is called Fmax in the literature (the F1-score at the threshold, which provides the maximum performance). Please use this terminology.
Answer:
Thank you for your comment, we have modified the terminology.
- "Each protein was annotated with its corresponding GO-terms from Uniprot using ViSEAGO"
It is not clear why this package is used? GO annotations of UniProt protein entries can directly be downloaded from UniProt or from the QuickGO browser. What is the use of ViSEAGO here?
Answer:
Since we performed major data analysis using R, we decided to utilize the existing Bioconductor tools to simplify downloading, parsing and transforming into needed format all GO annotations for UniProt. ViSEAGO package provides a simple interface to GO via a single function Uniprot2GO (https://rdrr.io/bioc/ViSEAGO/man/Uniprot2GO.html). This function downloads the current_release_numbers file (ftp://ftp.ebi.ac.uk/pub/databases/GO/goa/current_release_numbers.txt) directly from Uniprot-GOA. So functionally this is identical to manual downloading and parsing of GO annotations.
- " 2.7. Analysis of interactome profiles"
The analysis is not explained clearly. It sounds like this chapter talking about the method used for "3.6. Differences of splice-forms interactomic profiles", but there are sentences that sound irrelevant. For example: " Certain related groups were annotated by Gene Ontology [34] terms using GOrilla web service". What is the meaning of this? Why and how did you do this GO annotation? What is the significance of the results obtained from the GOrilla annotation?
Answer:
Thanks for this note. We rewrote this part of the manuscript for more clear understanding. Also we added Supplementary Figure S5 as an example of GO annotation by GOrilla.
- " Thus, the original binary 256x254 prediction matrix (proteins vs GO-terms) was reduced to 209 a 127x15 matrix.''
Why the number of proteins are reduced as well at the end of GO clustering process? According to the inheritance rule of GO DAG, a gene/protein annotated with a GO term is assumed to be annotated with the parents of that terms. As a result, the number of proteins should have remained the same.
Answer:
In order to adequately visualize function predictions we needed to reduce the number of both proteins and terms. For GO-terms we at first removed too rare terms (present in five or less proteins) and then removed too high-level terms (from each of 15 clusters of GO-terms selected the single least specific category having the lowest level). These two steps reduced the number of GO-terms from initial 204 to 15. As for the proteins we also removed low-annotated proteins (having five or less GO-terms), so initial 256 proteins were reduced to 127. The original phrase “binary 256x254 prediction matrix” is a typo and should be read as “binary 256x204 prediction matrix”. We thank the Reviewer for raising this issue. We have modified the manuscript according to this comment.
Reviewer 3 Report
Dear Authors,
This article is very difficult to read. It must be thoroughly reformatted to improve the understandability to readers.
Author Response
Dear Reviewer,
we are really sorry to hear that you have found the manuscript difficult to read, but we appreciate the opportunity to clarify our research objectives and results. We have significantly modified the paper structure and data presentation. We have redrawn and simplified several figures, added descriptions for the functions and packages used, and wrote sub-sections 3.7 and 3.8 to better explain how we compared our function predictions with existing data from UniProt. The list of cited literature has been enriched by more than 20 items - all of them are recent articles valuable for biochemists and bioinformatics working in the field of functional protein annotation. We tried our best to make it as clear as possible by thoroughly describing the proposed approach to protein function prediction. We believe that the improvements will increase the likelihood that our manuscript will be useful to other researchers.
Round 2
Reviewer 1 Report
The present version of the paper has improved and answer to most my concerns
Reviewer 2 Report
The authors addressed all of my issues.